# Investigation of NLR Genes Reveals Divergent Evolution on NLRome in Diploid and Polyploid Species in Genus *Trifolium*

**DOI:** 10.3390/genes14040867

**Published:** 2023-04-04

**Authors:** Amna Areej, Hummera Nawaz, Iqra Aslam, Muhammad Danial, Zohaib Qayyum, Usama Akhtar Rasool, Jehanzaib Asif, Afia Khalid, Saad Serfraz, Fozia Saleem, Muhammad Mubin, Muhammad Shoaib, Muhammad Shahnawaz-ul-Rehman, Nazia Nahid, Saad Alkahtani

**Affiliations:** 1Evolutionary Biology Lab, CABB, University of Agriculture, Faisalabad 38000, Pakistan; 2Department of Botany, Division of Science and Technology, University of Education, Lahore 55210, Pakistan; 3Metabolomics Innovative Institute, University of Alberta, Edmonton, AB T6G 2R3, Canada; 4Institute of Health Sciences Islamabad, Khyber Medical University, Peshawar 25000, Pakistan; 5Department of Biotechnology and Bioinformatics, Government College University, Faisalabad 54000, Pakistan; 6Department of Zoology, College of Science, King Saud University, P.O. Box 2455, Riyadh 11451, Saudi Arabia

**Keywords:** clover, novel resistance resources, NLR genes, phylogeny wild *Trifolium* species

## Abstract

Crop wild relatives contain a greater variety of phenotypic and genotypic diversity compared to their domesticated counterparts. *Trifolium* crop species have limited genetic diversity to cope with biotic and abiotic stresses due to artificial selection for consumer preferences. Here, we investigated the distribution and evolution of nucleotide-binding site leucine-rich repeat receptor (NLR) genes in the genus of *Trifolium* with the objective to identify reference NLR genes. We identified 412, 350, 306, 389 and 241 NLR genes were identified from *Trifolium. subterraneum*, *T. pratense*, *T. occidentale*, subgenome-A of *T. repens* and subgenome-B of *T. repens*, respectively. Phylogenetic and clustering analysis reveals seven sub-groups in genus *Trifolium*. Specific subgroups such as G4-CNL, CCG10-CNL and TIR-CNL show distinct duplication patterns in specific species, which suggests subgroup duplications that are the hallmarks of their divergent evolution. Furthermore, our results strongly suggest the overall expansion of NLR repertoire in *T. subterraneum* is due to gene duplication events and birth of gene families after speciation. Moreover, the NLRome of the allopolyploid species *T. repens* has evolved asymmetrically, with the subgenome -A showing expansion, while the subgenome-B underwent contraction. These findings provide crucial background data for comprehending NLR evolution in the *Fabaceae* family and offer a more comprehensive analysis of NLR genes as disease resistance genes.

## 1. Introduction

Clover plays a vital role in major agroecological zones as one of the most significant forage crops, and it serves as a crucial component for sustainable intensification of livestock farming systems [1]. It contains a high protein content and does not require considerable nitrogen fertilizer input unlike other forage crops that belong to the family *gramineae* [2]. Clover belongs to one of the largest genera *Trifolium* in the family *Fabaceae* and consists of ca. 255 species [1]. It occurs naturally in subtropical and temperate regions of both the Southern and Northern Hemispheres. Southeast Asia and Australia lack native clovers [3]. The genus *Trifolium* is believed to have originated from the Mediterranean basin during the early Miocene period (16–23 Mya) owing to the abundant species diversity, which encompasses a wide range of chromosome numbers and endemic species. East Africa and North America is considered as secondary center of distribution [3,4].

Unlike other agronomical crops, genus *Trifolium* consist of multiple cultivated species, major cultivated species being *T. hybridum* (alsike clover), *T. pratense* (red clover), *T. repens* (white clover) and *T. alexandrinum* (berseem) [4,5]. Sheer diversity in terms of ploidy and number of chromosomes have been reported from this genus. The basic chromosome number is 8, and the ancestral chromosomal composition is considered to be diploid with 2n = 16. Thirty-one species have been found to have identified aneuploid series of (x = 7, 6, 5) [4]. Polyploidy is considered as driver of speciation and biodiversity [6]. Here, white clover is a allopolyploid from the interspecific hybridization of two progenitor species *T. occidentale* and *T. pallescens* [7,8]. Polyploidy events can confer increased vitality, phenotypic and genotypic plasticity with improved adaptability. Both diploid progenitor species have limited habitats for example *T. occidentale* is confined to ~100 m of the western coastal region of Europe, and *T. pallescens* is restricted to European alpine habitats at higher altitudes. In contrast, the interspecific hybrids *T. repens* that were produced have a broad natural range covering grasslands in Europe, Western Asia and North Africa at different latitudes and altitudes [8]. Polyploidization events have been shown to provide an advantage in harsh environmental conditions, as demonstrated during the Cretaceous-Paleogene mass extinction [6,9,10,11,12]. Understanding the evolution of certain gene families in ancestral and polyploid species notifies the underlying mechanism responsible for their diversification. *Trifolium* cultivated species face a continuous threat from disease pathogens, for example, Alfalfa mosaic virus (AMV), Red clover vein mosaic virus (RCVMV), Aphanomyces root rot, and clover rot by fungal pathogens [13]. Therefore, gaining an understanding of the underlying molecular mechanism of disease resistance genes, their accurate identification and interpretation is vital for achieving higher production rates.

Direct or indirect interactions with a pathogen’s effector activate defensive mechanisms such as localized programmed cell death or hypersensitive response, by nucleotide-binding site leucine-rich repeat receptors (NLRs) [14]. NLRs consist of two main parts, the nucleotide-binding domain (NB-ARC) and the C-terminal leucine-rich repeats (NLRs), with the NB-ARC domain being the highly conserved segment used to determine evolutionary relationships between plant NLRs [15]. The plant NLRs are categorized into four significant classes, each with distinct N-terminal domain: TIR-NLR subclade with an N-terminal Toll/interleukin-1 receptor (TIR) domain, CC-NLR subclade with an N-terminal type Rx-type coiled coil (CC) domain, CCR-NLR subclade with RTP8-type CC domain, and G10 subclade, a newly proposed category with a unique type of CC, forming a monophyletic group. Genome-wide analysis of *Trifolium* species has not yet been conducted. It is a formidable undertaking to detect and mark NLR genes in plants across the entire genome as a result of their intricate sequence variability and evolutionary past. However, the functional annotation of plant resistance genes can be performed in a high-capacity approach using NLRtracker, a newly developed tool that utilizes the canonical features of these genes [16].

To date, genome and transcriptome resources are available for four species, *T. occidentale*, *T. repens*, *T. subterraneum* and *T. pratense.* One of the progenitors of white clover *T. pallescenes* genome is sequenced; however, its assembly genome is not available. The *Trifolium* genus presents a valuable opportunity to investigate the evolution of NLR genes in both diploid and tetraploid species. Our study involved identifying and characterizing NLR genes in three diploid and one tetraploid species of *Trifolium*. Our research aimed to address several complex questions, such as: What is the distribution of the NLR gene number and classes across the *Trifolium* genus? How does artificial selection affect the distribution of NLR genes? How does allopolyploidy impact NLR gene evolution in *Trifolium*? Finally, what are the primary evolutionary mechanisms employed by *Trifolium* wild relatives to enhance their genetic diversity and combat biotic and abiotic stress factors?

## 2. Materials and Methods

### 2.1. Mining of NLR Genes in Arachis Species

The NCBI genome database (Appendix A) was utilized to acquire the genome assembly for four *Trifolium* species (Appendix A). Augustus (v-3.4.0) was used to annotate these genomes using predefined configurations, with the exception of the full gene models option (genemodel = complete). A couple of Perl modules (getAnnotFasta.pl and gffread) were then utilized to obtain amino acid and coding sequences from the resulting gff file [17]. To facilitate the comparison between ancestral species, the tetraploid species *T. repens* was separated into individual subgenomes-A and B. The NLRtracker pipeline was applied to analyse the refrence proteomes of all the *Trifolium* species, which extract and annotate NLRs from transcript and protein file. The NLRtracker pipeline extracted NLRs and provided domain configuration dervied on the bases of features identified in reference plant NLR genes using Interproscan [18] and predetermined NLR motifs [19]. However, the manual curation was performed using using the clustering and phylogenetic analysis for each NLR gene, as the CCR-NLR annotation performed by NLRtracker was indeterminate.

### 2.2. Classification and Phylogenetic Analysis

The reference NLR genes from PRG database was used to produce a library of NB-ARC domains [15]. Clustering of these domains was accomplished through UCLUST [20] with an identity threshold of 50%, as previously mentioned [21]. Subsequently to the formation of each cluster, reference genes were classified into subgroups designated by Eunyoung Seo et al. [22] and used in phylogenetic and clustering research as seed probes. For an exhaustive phylogenetic investigation, using MUSCLE (version 1.26) the extracted NB-ARC domains from *Trifolium* species (output of NLRtracker) were aligned with the NB-ARC seed probes. Conducting the maximum likelihood analysis through IQtree v 2.0 [23], the most appropriate model of evolution (−m VT + F + R9) was chosen, with 1000 bootstrap replicates being incorporated in the process.

### 2.3. Loci Maps and Syntenic Maps of NLR-Genes

The NLR genes coordinates were analyzed using density distribution analyses. Scaffolds that were not placed were discarded, and binning was limited to solely chromosomal contigs. We utilized the bin size of 5 Kb for calling each gene within the genome using the BEDtools program [24], using the “make-windows” and “intersect” commands. Assigning a serial number to each bin was carried out manually dependent on its NLR density value. Visualizing a linearized version of the genome based on the bin number and NLR density value in each bin, the Rideogram package [25] was utilized. In order to investigate the syntenic relationship between NLRs in *Trifolium* species, genomic tracks were initialized using BED files from each species with a bin size of 5 Kb. Count file from BEDtools were subjected to BLAST analysis for the identification of inter-genomic similarities and sorted according to the BLAST output. Collinearity between genes was used to provide genomic linkage. Synteny plot were visualized using the R package “Circlize” [26].

### 2.4. Evolutionary Analysis in Trifolium NLRs

To align the nucleotide sequences of paralog groups, the concluded protein sequences were first aligned beyond their corresponding subgroups using Clustalw. The aligned nucleotide sequences were obtained using pal2nal, a Perl-based software with gaps, and N-coding codons subsequently removed [27]. The KA/KS calculator was employed with the MA method to estimate KS values, and only significant duplication events were retained by conducting the Fisher test (F-test) on each paralog selection value (with *p* value > 0.01). This approach ensured uniqueness while maintaining the word count. Ks-values greater than two (>2) were excluded from further analysis due to the possibility of substitution saturation [28]. Finally, Ortho-venn2 was applied to cluster NLR genes and study their orthologs [29]. Using default settings and an E-value of 10^−2^, the Orthovenn2 program was used to query putative NLR genes from each species. All identified NLR genes were subjected to Orthofinder for orthology analysis [30]. The resulting orthogroups were manually labeled, and R package APE [31] was utilized for building an ultrametric tree. Both the orthogroups and ultrametric tree files were used as input for CAFE5 [32], and gene gain and loss at each node was performed by parsing the resulting files. Ortholog sequences between the A and B genomes of *T. repens* and their ancestral sequences *T. occidentale* were also obtained from Orthofinder [30].

### 2.5. RNA-Seq Based Expression Analysis

Basal expression level of NLRs determined from this study was analyzed using the available datasets of *Trifolium* and its related species. First dataset provides extensive collection of replicates from root tissue, emerging leaf, first mature leaf, 3–4 nodes and young open florets (PRJNA521254 and PRJNA523044). The alignment of raw read sequences was conducted in this study using HISAT [33] and the reference genome (v.2). Subsequently to the generation of alignments, StringTie [33] was employed for the assembly of transcripts. To group experimental conditions and determine differential expression between them, Ballgown [33] was used to process the assembled transcripts and their corresponding abundance.

## 3. Results

### 3.1. Gene Mining of NLR Genes in Trifolium Species

Here, we utilized the NLR tracker pipeline [16] for NLR genes mining and successive annotations. In total 412, 350, 306, 389 and 241 NLR genes were identified from *T. subterraneum*, *T. pratense*, *T. occidentale*, subgenome-A of *T. repens* and subgenome-B of *T. repens*, respectively (Figure 1). All four major classes of NLR gene families were present in all four species, with subclass CC-NLR exhibiting the highest contribution and CC_R_-NLR showing the least number of homologs, which was consistent with the previous reports [21]. Overall, an apparent correlation was not observed between the genome size and number of NLR genes. *T. subterraneum* has shown largest expansion of NLR gene family even though it has a genome size of 460 Mb as compared to subgenome-A of *T. repens* (513 Mb) and *T. occidentale* (484 Mb). The allotetraploid *T. repens* interestingly shows an expansion of the NLR gene family in subgenome-A as compared to its ancestor *T. occidentale*. It implies that genome expansion also favored the duplication of NLR gene family. CC_R_-NLR is the most conserved class of NLR genes, and its repertoire remains stable across *Fabaceae* members [21]; however, the subgenome-A of *T. repens* have shown their significant expansion. It is consistent with the previous observation that polyploidization may increase or decrease the number of certain genes families [34]. The overall pattern of NB-ARC encoding genes in all *Trifolium* species is CNL of 46.5% and TIR-NLR up to 38%. The repertoire of NLR sequence identified in this study is provided in the Appendix A.

### 3.2. Landscape of NLR Genes among Trifolium Species

We further studied the landscape of NLR genes in three genomes, *T. pratense*, *T. occidentale* and two subgenomes of *T. repens*, by plotting the gene density of NLR genes on linearized chromosomes (Figure 2B). Major NLR gene clusters were observed in chromosomes 1, 2, 4, 5 and 6 across all three species. Interestingly *T. pratense* shows the highest gene density with respect to its size. We also observed the NLRome of subgenome-A of *T. repens* have shown expansion as compared to its progenitor *T. occidentale*. Furthermore, we compared the syntenic relationship between these three species (Figure 2A). A significant degree of syntenic connections were identified between three species. Highly conserved homeologous clusters were identified in the syntenic comparison between subgenome-A and -B of *T. repens*. In addition, conserved clusters of NLR genes were identified on chromosome 1, 2, 5 and 6 between subgenome A and its progenitor *T. occidentale*. In addition, subgenome-B also showed the syntenic conservation with *T. occidentale*.

### 3.3. Phylogenetic Analysis and Classification of NLR Genes

The classification of the conserved NB-ARC domain sequence was performed by grouping at 75 percent identity using CD-HIT. Representative members of each cluster were utilized for the contraction of phylogenetic relationship among *T. repens* (A and B sugenome), *T. pratense, T. occidentale* and *T. subterraneum* (Figure 3). In addition, reference NLR genes were also included for the classification of NLR genes in genus *Trifolium.* TNL clade was branched out as expected; however, five major radiations were observed. Class CNL was divided into three major subclades: CC_R_-NLR, CC-NLR and CC_G10_-NLR. Significant divergence was observed for subgroups of CC-NLR including G4-CNL, G7-CNL, CNL-Un and CNL-G11. Consistently with previous observation from *Fabaceae* members [21], CNL groups G1-G8 were also absent in this genus, *Trifolium.* This conclusion strongly suggests the hypothesis that all *Fabaceae* members lack G1-G8 subgroups (Figure 3).

Phylogenetic analysis further suggest that species such as *T. pratense* and *T. subterraneum* substantially gained a CC_G10_-CNL subclass of genes as compared to *T. repens* and its progenitor. On the other hand, *T. repens* followed by an alloploidization event have substantially gained G4 and G7, as significant diversification can be observed in both subgroup clades. Interestingly, *T. subterraneum* has the highest number of TIR and CC-NLR genes considering its diploid nature. These unbalanced gene duplication occurrences across *Trifolium* genus suggest the possible role of terminal duplication after the divergence from common ancestor.

We also compared the selection pressure within in the pairs of paralogs from four major subgroups (G4, G7, CCG10-NLR and TIR-NLR). G4 and G7 have *Ka/Ks* above 0.5 in *T. subterraneum* and *T. pratense* as compared to other species, *T. occidentale* and *T. repens* (A and B), where its value is less than value 0.5. It suggests that the tendency of expansion of G4 and G7 is higher in *T. subterraneum* and *T. pratense*. This observation is consistent with the fact that preferential expansion of G4 and G7 is also observed in other *Fabacaeae* genera such as *Dalbergia* [21] (Appendix A).

### 3.4. NLR Gene Birth and Death Ratio

Orthologs analysis revealed that, in total, 52 core ortholog clusters are present in these species. Gene families evolve either by birth and death of gene families or duplication of the number of genes within these families. Here, Orthofinder and CAFÉ analysis shows the birth and death of gene families (green/red) and number of genes duplicated (blue) in Figure 4. We constructed the phylogenetic tree for each species and its subgenome with birth and death of genetic events among members of the genus *Trifolium*. *Lotus japonicus* was considered as the most related outgroup for the common ancestor of vicioid clade. This analysis revealed that net contraction of NLR gene family occurred; however, large scale terminal duplication was observed within gene families in Lotus japonica. Common ancestor of genus *Trifolium* suggest loss of NLR gene families, which is consistent with overall contraction during diploidization after one common whole-genome duplication of all *Fabaceae*, nearly 58.5 Mya [22]. Similarly, the trend of contraction continued for cultivated species of *T. occidentale*, where a net death of gene families occurred with reduced terminal duplication of NLR genes. After the alloploidic event of *T. repens*, both subgenomes have shown divergent evolution of NLR gene, and significant loss of NLR gene families occurred in subgenome-B of *T. repens* in contrast to the substantial gain of NLR gene families in subgenome-A. Similarly, a marked increase in gene duplication was also observed for subgenome-A as compared to subgenome-B. Other species have shown a relative expansion of NLRome. *T. pratense* have shown an overall loss of gene families, but a considerable number of genes were gained due to terminal duplication after speciation. Similarly, *T. subterraneum* have shown the highest expansion of NLRome by birth of gene families and terminal duplication within these gene families. It is consistent with values obtained from ka/ks that suggests analysis of a higher selection rate on *T. subterraneum* and *T. pratense* as compared to other species (Appendix A).

### 3.5. Duplication Assay

Divergent evolution of NLRome of genus *Trifolium* could possibly be due to the multiple evolutionary mechanism including duplication, transposition and recombination. Here, we explored the duplication history of genus *Trifolium* by comparing the Ks values between paralogs each subgroups (Figure 5). Notably the *Trifolium* lineage shows rapid mutation rate of 1.8 × 10^−8^ per base per generation similar to genus Arachis [8]. The divergence of four common ancestor available species, *T. repens*, *T. occidentale*, *T. pratense* and *T. subterraneum*, occurred during approximately 13 Mya [35]. Collective Ks values obtained from all NLR subgroups suggest one common duplication curve between 0.12 to 0.24 during the timeline of 3.3 to 6 Mya. It strongly suggests that a major duplication of NLRome in *Trifolium* species occurred after speciation from their common ancestor. Subgroups G4-CNL, G7-CNL and TIR-NLR have been amplified dramatically through gene duplication events after speciation. Interestingly, a marked increase in duplicated pairs can be observed for G10-CNL in *T. pratense* and *T. subterraneum*, thus suggesting the species-specific diversity of NLRome. In the case of allopolyploid *T. repens*, subgenome-A and its progenitor have shown the highest frequency of gene duplication in contrast to subgenome-B.

### 3.6. Expression of Identified NLR Genes in Trifolium Species

We further evaluated the expression of identified NLR genes in two species of *T. occidentale* and *T. repens* (A and B subgenomes). Expression was five different stages of plant development including emerging leaf, first mature leaf, nodes 3,4, root tissue and young open florets. Higher qualitative and quantitative expression of NLR genes was observed in *T. occidentale* (PRJNA521524), and in total, 49 genes have shown significant expression. The majority of these genes belong to G4, -G7 and G10 subgroups (Appendix A). Secondly, we also compared the expression of NLR genes for two subgenomes of *T. repens* in the same type of tissue (A and B). Overall, 29 genes NLR genes were expressed in five different tissue types (mentioned earlier) from subgenome-A (progenitor: *T. occidentale*), and 23 genes were NLR genes expressed from subgenome-B (progenitor: *T. pallenscens*). In the case of subgenome-A, four genes (g233455, g222, g142003 and g17216) have shown higher expression in root tissues and the emergence of first mature leaf (Figure 6), whereas two genes (g7060 and g142203) from the subgenome-B of *T. repens* have shown the higher expression in root tissues and young open florets. Overall, these analyses suggest that NLR genes were expressing in variable quantitative and qualitative fashion in both subgenomes. Additionally, progenitor species *T. occidentale* have shown the highest expression of NLR genes during different stages of development.

## 4. Discussion

It is consistent with the fact that each plant lineage is under consistent battle against different types of pathogens, thus developing a unique repertoire of NLR genes [16,36]. Here, we studied the evolution of NLR genes in four species of genus *Trifolium*, which suggests they are evolving in species-specific divergent evolution. *Trifolium* species have shown a continuous trend of NLRome contraction consistent with global contraction after diploidization; however, certain species such as *T. subterraneum* have shown dynamic expansion due to gene duplication and birth of gene families. It is common knowledge that gene duplication produces valuable paralogs that can provide necessary genotypic variability. Because of the NLR duplication, there is a possibility that recombinant NLR genes may develop, which will have unique activities and expression patterns that will aid in the fight against infections [37]. The comparison of NLR across members of the genus revealed that some subgroups underwent restriction and selective expansion as a result of asymmetrical gene duplication occurrences. Subgroups G4-CNL and G7-CNL have shown substantial diversification due to their large-scale terminal duplication indicated by gene birth and death analysis. In the case of *T. pratense*, large-scale terminal duplication after speciation facilitated the diversification of CC_G10_-NLR between 4 and 6 Mya.

Polyploidy and aneuploidy provides a new perspective towards understanding NLR genome evolution during swift genomic rearrangements. *T. repens* is the successful allotetraploid naturalized globally; by contrast, its diploid extant progenitor remains in an extreme specialized habitat. It has enabled niche expansion, which has facilitated global radiation of previously confined specialist progenitor genomes [8]. This occurred in the past ~20 Kya during the last glaciation, when alpine and coastal species were likely in proximity. It should be noted that major findings cannot be drawn for the subgenome-B evolution due to the unavailability of the assembly of its progenitor *T. pallescens* genome. Considering the fact that progenitor genome integrity has been retained in *T. repens* [8], we can hypothesize that both subgenomes are evolving in an asymmetrical manner. Our analysis suggests that both subgenomes A and B are evolving in a divergent manner, where NLRome of subgenome-A is undergoing continuous expansion, whereas B-NLRome is likely contracting. Genome rearrangements and loss of duplication genes are common feature of polyploids as they progress toward a diploidization [8]. Recent whole-genome duplication favoured the expansion of subgroups G4-NLR, G7-NLR and especially CC_R_-NLR (helper-NLR), which have shown significant expansion, unlike any other counterpart in the family *Fabaceae*. Furthermore, homeologous sequence exchanges (HSEs) between subgenomes play an important role in gene dosage due to chromosomal rearrangement. In future, HSEs should be studied in detail toward understanding the expansion of *T. repens.* In addition, structural variation (SVs) also play a pivotal role in the evolution of various gene families across different polyploids, e.g., cotton and brassica [34,38]. In future, we will explore the role of SVs on the evolution of NLR genes and its related families in genus *Trifolium.*

## 5. Conclusions

Genus *Trifolium* provides an interesting example to study the evolution of NLRome, since it has multiple cultivated and wild species as well as diploid and polyploid species. Our comprehensive results indicate that specific subgroups of NLR genes have shown different duplication patterns in specific species, which strongly suggests that subgroup duplication is sign of their differentiation and evolution. Each plant species of *Trifolium* genus is consistently combating different type of pathogens and hence developing a unique NLR gene library, as *T. subterraneum* have shown the expanded NLRome among all five species of *Trifolium.* In addition, our results strongly suggest the asymmetric evolution of NLRome in subgenomes-A and B of the allotetraploid species of *T. repens*. Genome and transcriptome sequencing of divergent cultivated and wild species will further elucidate the effect of natural and artificial selection on the evolution of NLRome in this important genus.

## Figures and Tables

**Figure 1 genes-14-00867-f001:**
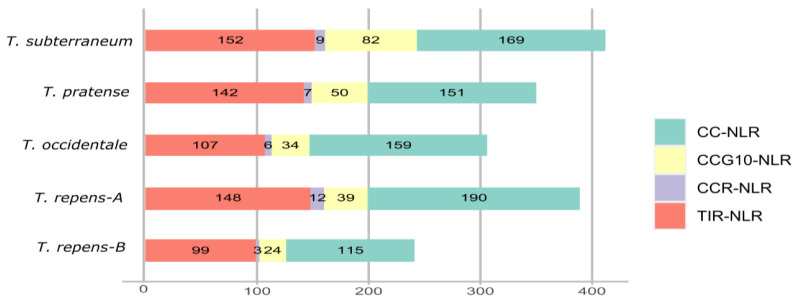
Horizontal bar graph exemplifies the organization of NLR genes that include four significant sub-class denoted in four different colors.

**Figure 2 genes-14-00867-f002:**
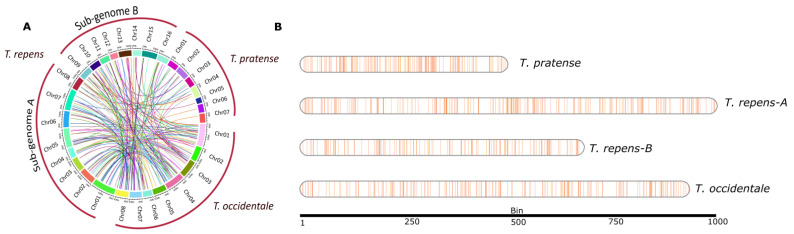
Syntenic and gene density plots of *Trifolium* species. (**A**) Synteny analysis of NLR genes between *T. repens* (A and B subgenomes), *T. occidentale* (progenitor) and *T. pratense.* (**B**) The NLR genes located on chromosomes are denoted in vertical blue and orange lines, inferring the gene density map between genomes assemblies of three *Trifolium* species.

**Figure 3 genes-14-00867-f003:**
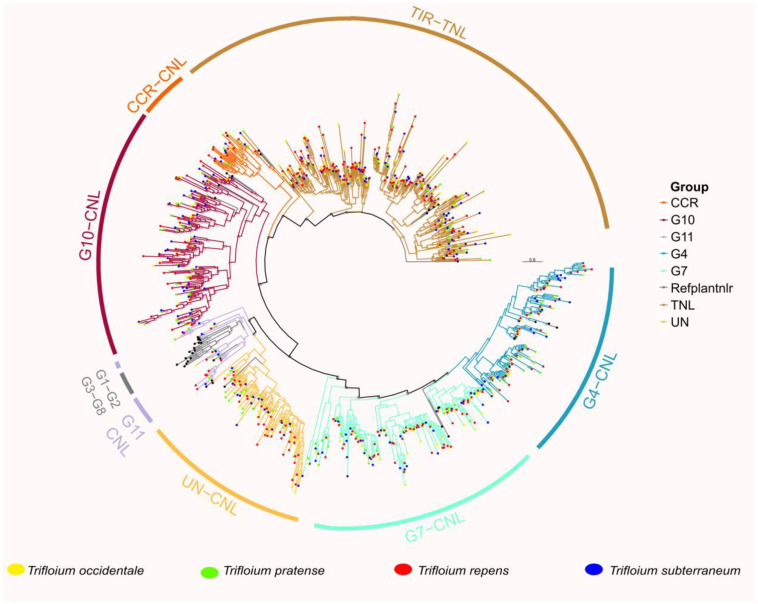
Phylogenetic reconstruction of NLR genes on the basis of conserved domain NB-ARC. Seven subgroups of NLR genes were identified in four species: *T. occidentale*, *T. pratense*, *T. repens* and *T. subterraneum*.

**Figure 4 genes-14-00867-f004:**
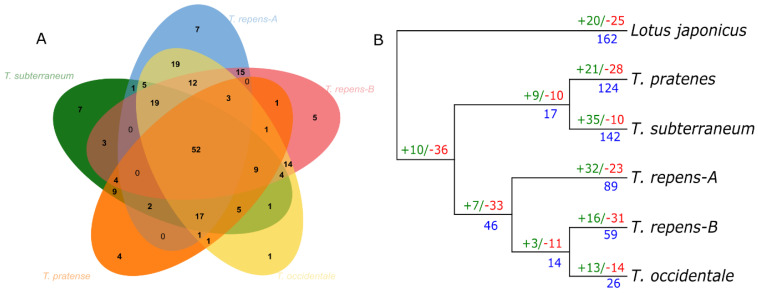
The number of gene gain and gene loss analysis. (**A**) The ortho Venn diagram depicts the shared NLR genes among species of *Trifolium*. (**B**) The red color code represents the gene loss, green indicates gene gain and blue denotes gene duplication.

**Figure 5 genes-14-00867-f005:**
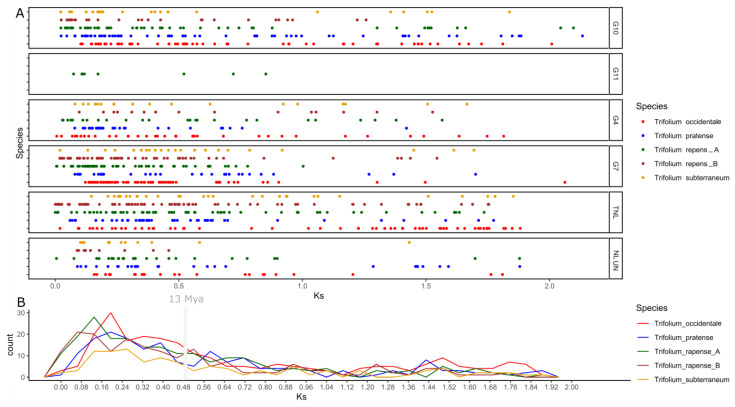
Estimation of NLRome duplication event in the genus *Trifolium*. All five species belonging to the genus *Trifolium* with Ka/Ks values of their paralogs are denoted. (**A**) The vertical axis indicate Ks value and horizontal axis shows frequencies. (**B**) The generalized observed duplication manner in all species.

**Figure 6 genes-14-00867-f006:**
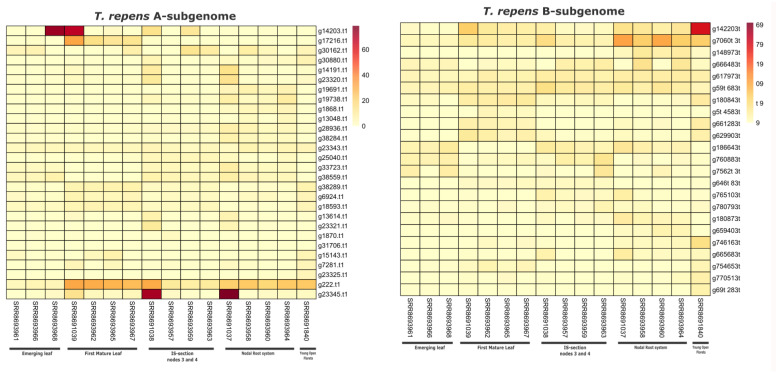
Comparative transcriptomic analysis between subgenomes–A and B of *T. repens* during five different development stages. Left panel shows the expression of 29 different NLR genes in subgenome-A, whereas right panel shows the expression of 23 NLR genes in subgenome-B of *T. repens*.

## Data Availability

All the accession numbers utilized in this study are provided. NLR genes identified in this study are included in the Appendix A.

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
