# Peer review of "Investigation of NLR Genes Reveals Divergent Evolution on NLRome in Diploid and Polyploid Species in Genus Trifolium"

_genes, 2023, doi:10.3390/genes14040867_

Round 1

Reviewer 1 Report

To,

The Editor,

Genes, MDPI,

Manuscript ID: genes-2294905

Subject: Submission of comments of the manuscript in “Genes"

Dear Editor Genes, MDPI,

Thank you very much for the invitation to consider a potential reviewer for the manuscript (ID: genes-2294905). My comments responses are furnished below as per each reviewer’s comments. 

In the reviewed manuscript, the authors investigated the distribution and evolution of nucleotide binding site leucine rich repeat receptor (NLR) genes in the genus of Trifolium with the objective to identify reference NLR genes. We identified 412, 350, 306, 389 and 241 NLR genes were identified from T. subterraneum, T. pratense, T. occidentale, subgenome-A of T. repens and subgenome-B of T. repens, respectively. Phylogenetic and clustering analysis reveals 7 sub-groups in genus Trifolium. Specific subgroups like G4-CNL, CCG10-CNL and TIR-CNL show distinct duplication patterns in specific species, suggesting subgroup duplications that is the hallmarks of their divergent evolution. Furthermore, our results strongly suggest the overall expansion of NLR repertoire in T. subterraneum due to gene duplication events and birth of gene families after speciation. NLRome of allopolyploid species T. repens have been evolving in an asymmetrical manner as A-subgenome have shown expansion while NLRome of B-subgenome has underwent contraction. These results will allow a more thorough analysis of NLR genes (disease resistance genes) and give crucial background data to comprehend NLR evolution in the Fabaceae family. There are some minor and major comments. The author should revise as per my comments and suggestions. 

  1. I have read the entire manuscript and my initial comment is that manuscript is poorly written. I have significant concerns about the grammar and vocabulary of the manuscript; therefore, I recommend the authors to use an English proofreading service.
  2. The abstract does not reflect the whole story, revise it
  3. The key words must be in alphabetical order.
  4. The writing style of the paper is very poor. There are many grammatical mistakes. Long sentences with noticeable grammatical mistakes are frequently present throughout the manuscript. There are many typos mistakes in this whole manuscript. The author should check the whole manuscript.
  5. The introduction part is not impressive and systematic. In the introduction part, the authors should elaborate on the scientific issues in plant research. The Content of the introduction is effective in essence but very poorly presented, significant improvements are needed in presenting the proper background of the work undertaken
  6. The figures are quite low resolution and difficult to make out. Higher-resolution versions will be needed for publication. Further, text in figure is not readable, for example, in Figures 1, 2, 3, 4A, 5, and 6.
  7. The discussion should be interpreted with the results as well as discussed in relation to the present literature. 
  8. Authors must add the conclusion section in the manuscript.

Author Response

Response to comments made by reviewer 1

The responses in the final draft for reviewer 1 are highlighted in green colour

  1. I have read the entire manuscript and my initial comment is that manuscript is poorly written. I have significant concerns about the grammar and vocabulary of the manuscript; therefore, I recommend the authors to use an English proofreading service.

>> We sincerely thank the reviewer for providing the important comments for the improvements of this manuscript.

  1. The abstract does not reflect the whole story, revise it

>> The abstract is substantially revised and modified form can be seen in the final draft

  1. The key words must be in alphabetical order.

>>  The changes were completed as suggested by the reviewer.

  1. The writing style of the paper is very poor. There are many grammatical mistakes. Long sentences with noticeable grammatical mistakes are frequently present throughout the manuscript. There are many typos mistakes in this whole manuscript. The author should check the whole manuscript.

>> Frequent checks have been performed to remove any grammatical mistakes found in the MS.

  1. The introduction part is not impressive and systematic. In the introduction part, the authors should elaborate on the scientific issues in plant research. The Content of the introduction is effective in essence but very poorly presented, significant improvements are needed in presenting the proper background of the work undertaken

>> Several parts of the introduction is being restructured and rewritten to make this part more effective and understandable to general audience.

  1. The figures are quite low resolution and difficult to make out. Higher-resolution versions will be needed for publication. Further, text in figure is not readable, for example, in Figures 1, 2, 3, 4A, 5, and 6.

>> High resolution figures are now provided in the portal where clear representation of minor details are provided.

  1. The discussion should be interpreted with the results as well as discussed in relation to the present literature. 

>> The discussion part is also improved however there was little room available as the data in Trifolium species is still very limited.

  1. Authors must add the conclusion section in the manuscript.

>> Conclusion part is now added in the MS.

Reviewer 2 Report

The manuscript " Investigation of NLR genes reveals divergent evolution on NLRome in diploid and polyploid species in genus Trifolium " indicated that specific subgroups show different duplication patterns in specific species, suggesting that subgroup duplication is a sign of their differentiation and evolution. The authors reveal that each plant lineage is constantly struggling with different types of pathogens, thereby developing a unique NLR gene library. However, this manuscript may be accepted after minor modification.

Minor:

1. Page 1 Line 23-24, "suggesting sub-group duplications that is the hallmarks of their divergent evolution." should be "this suggests sub-group duplications that is the hallmarks of their divergent evolution."

2. Page 1 Line 37, "other forage crops that belongs to family gramineae " should be "other forage crops that belong to family gramineae."

3. Page 1 Line 38, " consist of ca. 255 species " should be "consists of ca. 255 species"

4. Page 2 LIne 78-79, " To this date no genome wide analysis on Trifolium species is not reported " should be "To this date no genome wide analysis on Trifolium species is reported" or "To this date genome wide analysis on Trifolium species is not reported" 

Author Response

Response to the comments made by Reviewer 2

On the behalf of all the author, I sincerely thank the reviewer 2 for your positive criticism and acknowledging the efforts made to under the evolution NLR genes in understudied crop species like Trifolium.

Indicated in blue color

Minor:

  1. Page 1 Line 23-24, "suggesting sub-group duplications that is the hallmarks of their divergent evolution." should be "this suggests sub-group duplications that is the hallmarks of their divergent evolution."

 >>> That’s rightly pointed out mistake, Its now corrected

  1. Page 1 Line 37, "other forage crops that belongs to family gramineae" should be "other forage crops that belong to family gramineae."

 >>>  The mistake is now corrected.

  1. Page 1 Line 38, " consist of ca. 255 species " should be "consists of ca. 255 species"

 >>>  The mistake is now corrected.

  1. Page 2 LIne 78-79, " To this date no genome wide analysis on Trifolium species is not reported " should be "To this date no genome wide analysis on Trifolium species is reported" or "To this date genome wide analysis on Trifolium species is not reported" 

>>> The mistake is now corrected

Reviewer 3 Report

Comments and suggestions for the authors

This study presents an interesting objective and is designed to meet the aims of the project. However, the English in the present manuscript is not of publication quality and major improvement is required to enhance clarity. I provide a few of my comments for the authors to answer them.

1. Part 2.1 of the manuscript is titled as "Mining of NLR genes in Arachis species". Should Arachis be Trifolium instead? 

2. In part 2.4 of the manuscript, it is noted that gaps has been removed before calculating Ks. Did you delete non-triplet indels or the entire gaps? Did you notice frameshift mutations, stop codons or other errors? please provide sufficient information about what you did and clearly state the reason.

3. Where are the bootstrap values mentioned in part 2.2 presented? The values should be incorporated in the the Maximum likelihood tree and the corresponding figure must be referenced in the appropriate section in Results.

4. All figures should be referenced in the text. Figures 2, 3, 4 and 5 are not mentioned in the text.

5. I suggest adding a conclusion section to the manuscript, so that the questions raised by the authors especially the effect of artificial selection on distribution of NLR genes as well as its impact against pathogens are concisely concluded.

6. English editing is suggested to correct a lot of grammatical and punctuation issues throughout the manuscript.Comments and suggestions for the authors

This study presents an interesting objective and is designed to meet the aims of the project. However, the English in the present manuscript is not of publication quality and major improvement is required to enhance clarity. I provide a few of my comments for the authors to answer them.

1. Part 2.1 of the manuscript is titled as "Mining of NLR genes in Arachis species". Should Arachis be Trifolium instead? 

2. In part 2.4 of the manuscript, it is noted that gaps has been removed before calculating Ks. Did you delete non-triplet indels or the entire gaps? Did you notice frameshift mutations, stop codons or other errors? please provide sufficient information about what you did and clearly state the reason.

3. Where are the bootstrap values mentioned in part 2.2 presented? The values should be incorporated in the the Maximum likelihood tree and the corresponding figure must be referenced in the appropriate section in Results.

4. All figures should be referenced in the text. Figures 2, 3, 4 and 5 are not mentioned in the text.

5. I suggest adding a conclusion section to the manuscript, so that the questions raised by the authors especially the effect of artificial selection on distribution of NLR genes as well as its impact against pathogens are concisely concluded.

6. English editing is suggested to correct a lot of grammatical and punctuation issues throughout the manuscript.Comments and suggestions for the authors

This study presents an interesting objective and is designed to meet the aims of the project. However, the English in the present manuscript is not of publication quality and major improvement is required to enhance clarity. I provide a few of my comments for the authors to answer them.

1. Part 2.1 of the manuscript is titled as "Mining of NLR genes in Arachis species". Should Arachis be Trifolium instead? 

2. In part 2.4 of the manuscript, it is noted that gaps has been removed before calculating Ks. Did you delete non-triplet indels or the entire gaps? Did you notice frameshift mutations, stop codons or other errors? please provide sufficient information about what you did and clearly state the reason.

3. Where are the bootstrap values mentioned in part 2.2 presented? The values should be incorporated in the the Maximum likelihood tree and the corresponding figure must be referenced in the appropriate section in Results.

4. All figures should be referenced in the text. Figures 2, 3, 4 and 5 are not mentioned in the text.

5. I suggest adding a conclusion section to the manuscript, so that the questions raised by the authors especially the effect of artificial selection on distribution of NLR genes as well as its impact against pathogens are concisely concluded.

6. English editing is suggested to correct a lot of grammatical and punctuation issues throughout the manuscript.Comments and suggestions for the authors

This study presents an interesting objective and is designed to meet the aims of the project. However, the English in the present manuscript is not of publication quality and major improvement is required to enhance clarity. I provide a few of my comments for the authors to answer them.

1. Part 2.1 of the manuscript is titled as "Mining of NLR genes in Arachis species". Should Arachis be Trifolium instead? 

2. In part 2.4 of the manuscript, it is noted that gaps has been removed before calculating Ks. Did you delete non-triplet indels or the entire gaps? Did you notice frameshift mutations, stop codons or other errors? please provide sufficient information about what you did and clearly state the reason.

3. Where are the bootstrap values mentioned in part 2.2 presented? The values should be incorporated in the the Maximum likelihood tree and the corresponding figure must be referenced in the appropriate section in Results.

4. All figures should be referenced in the text. Figures 2, 3, 4 and 5 are not mentioned in the text.

5. I suggest adding a conclusion section to the manuscript, so that the questions raised by the authors especially the effect of artificial selection on distribution of NLR genes as well as its impact against pathogens are concisely concluded.

6. English editing is suggested to correct a lot of grammatical and punctuation issues throughout the manuscript.

Author Response

Response to the comments made by reviewer 3

The corrections for reviewer 3 are highlighted in grey colour in the modified version of MS.

This study presents an interesting objective and is designed to meet the aims of the project. However, the English in the present manuscript is not of publication quality and major improvement is required to enhance clarity. I provide a few of my comments for the authors to answer them.

>> I sincerely thank the reviewer for his/her well thought-out comments. These comments further enabled us to refine our understanding towards measure of nucleotide substitution and artificial selection.

  1. Part 2.1 of the manuscript is titled as "Mining of NLR genes in Arachis species". Should Arachis be Trifolium instead? 

>>> Thanks for the pointing out the important mistake, it now being rectified in the final draft.

  1. In part 2.4 of the manuscript, it is noted that gaps has been removed before calculating Ks. Did you delete non-triplet indels or the entire gaps? Did you notice frameshift mutations, stop codons or other errors? please provide sufficient information about what you did and clearly state the reason.

>> Thanks for your comments, Clearly the gaps were removed before calculating ks. The frameshifts were quite rare and also we have not noticed so far stop codons within the codon-based alignments. Initially it was surprise for my to not observed frameshifts and stop codons in this gene family especially from Trifolium spp. Since most codons are under selective pressure to encode for an amino acid, stop codons have a different function and are not subject to this same pressure. As a result, stop codons may be more prone to random mutations and other evolutionary processes that could obscure their informative value.

  1. Where are the bootstrap values mentioned in part 2.2 presented? The values should be incorporated in the the Maximum likelihood tree and the corresponding figure must be referenced in the appropriate section in Results.

>>>Thanks for the comment, The corresponding figure is now appropriately reference in the result section. Secondly the bootstrap values are labeled in the phylogenetic tree, however due to smaller font size its not visible in the embedded figure 3 in this draft. The high resolution figure (> 300 dpi) clearly shows the bootstrap values at each node.

  1. All figures should be referenced in the text. Figures 2, 3, 4 and 5 are not mentioned in the text.

>> Thanks for pointing out the important error. This error is now rectified and all the figures are now referenced accordingly.

  1. I suggest adding a conclusion section to the manuscript, so that the questions raised by the authors especially the effect of artificial selection on distribution of NLR genes as well as its impact against pathogens are concisely concluded.

>> The conclusion section is now added in the modified version.

  1. English editing is suggested to correct a lot of grammatical and punctuation issues throughout the manuscript.Comments and suggestions for the authors

>> We tried our level best to improve and rectify grammatical and punctuation errors. Hopefully these changes will massively help in the improvement of language.
